# Acute Severe Heart Failure Reduces Heart Rate Variability: An Experimental Study in a Porcine Model

**DOI:** 10.3390/ijms24010493

**Published:** 2022-12-28

**Authors:** Jan Naar, Mikulas Mlcek, Andreas Kruger, Dagmar Vondrakova, Marek Janotka, Michaela Popkova, Otomar Kittnar, Petr Neuzil, Petr Ostadal

**Affiliations:** 1Department of Cardiology, Na Homolce Hospital, 150 30 Prague, Czech Republic; 2Department of Physiology, First Faculty of Medicine, Charles University, 128 00 Prague, Czech Republic

**Keywords:** acute heart failure, experimental model, heart rate variability, pig

## Abstract

There are substantial differences in autonomic nervous system activation among heart (cardiac) failure (CF) patients. The effect of acute CF on autonomic function has not been well explored. The aim of our study was to assess the effect of experimental acute CF on heart rate variability (HRV). Twenty-four female pigs with a mean body weight of 45 kg were used. Acute severe CF was induced by global myocardial hypoxia. In each subject, two 5-min electrocardiogram segments were analyzed and compared: before the induction of myocardial hypoxia and >60 min after the development of severe CF. HRV was assessed by time-domain, frequency-domain and nonlinear analytic methods. The induction of acute CF led to a significant decrease in cardiac output, left ventricular ejection fraction and an increase in heart rate. The development of acute CF was associated with a significant reduction in the standard deviation of intervals between normal beats (50.8 [20.5–88.1] ms versus 5.9 [2.4–11.7] ms, *p* < 0.001). Uniform HRV reduction was also observed in other time-domain and major nonlinear analytic methods. Similarly, frequency-domain HRV parameters were significantly changed. Acute severe CF induced by global myocardial hypoxia is associated with a significant reduction in HRV.

## 1. Introduction

Neurohumoral activation, including sympathetic overactivity and vagal tone withdrawal, plays an important role in the pathogenesis and progression of heart failure with reduced ejection fraction (HFrEF) [1,2]. Pharmacotherapy that aims to reduce renin-angiotensin-aldosterone axis activation and autonomic nervous system imbalance continues to be the cornerstone of therapy [3,4,5]. Nevertheless, the prognosis of HFrEF remains poor [6]. Recently, several neuromodulation strategies have been proposed in HFrEF patients; they directly target the autonomic nervous system to improve residual autonomic imbalance. Despite promising preclinical results, neuromodulation has failed to prove beneficial in randomized clinical trials [7,8,9].

Heart rate variability (HRV) is a parameter that reflects autonomic nervous system activation. It has been shown that therapeutic approaches that decrease mortality in HFrEF patients (physical activity, pharmacotherapy, cardiac resynchronization therapy) also increase HRV [10,11,12,13]. Thus, HRV improvement may be a desirable therapeutic aim of neuromodulation in the heart (cardiac) failure (CF) population. There are, however, substantial differences in HRV among CF patients. Some CF patients, who are characterized by a lower HRV, reflecting higher cardiac sympathetic nervous activity and more severe CF, may benefit from neuromodulation therapy [14]. The failure of neuromodulation therapy in clinical settings due to the enrollment of subjects with insufficiently advanced disease is debatable. Clinical data related to HRV in acute CF are scarce [15], and the effect of acute severe CF on HRV has not been experimentally explored in detail before.

The main objective of this study was to test the hypothesis that acute severe experimental CF significantly decreases HRV. We further aimed to describe individual differences in reaction to acute CF development using a broad range of HRV metrics.

## 2. Results

Electrocardiogram (ECG) data from 23 subjects were eligible for HRV analysis. One subject with continuous arrhythmia was excluded. The development of severe acute CF resulted in a significant decrease in systemic arterial blood pressure, cardiac output, and left ventricular ejection fraction, while heart rate significantly increased. Hemodynamic details related to the development of acute CF are outlined in Table 1.

### 2.1. Effect of Experimental Acute Heart Failure on Time-Domain Parameters of Heart Rate Variability

Acute CF led to a prominent and uniform decrease in the standard deviation of intervals between normal beats (SDNN; 50.8 [20.5–88.1] ms versus 5.9 [2.4–11.7] ms, *p* < 0.001), as well as the square root of the mean of the sum of the squares of differences between adjacent NN intervals (RMSSD; 46.3 [10.1–87.2] ms versus 2.0 [21.8–6.2] ms, *p* < 0.001) and a number of pairs of adjacent NN intervals differing by more than 50 ms divided by the total number of all NN intervals (pNN50; 4.0 [0.0–15.2]% versus 0.0 [0.0–0.0]%, *p* < 0.001; Figure 1). A representative example of time-domain HRV analysis is displayed in Figure 2.

### 2.2. Effect of Experimental Acute Heart Failure on Frequency-Domain Parameters of Heart Rate Variability

Spectral frequency analysis of HRV also revealed significant changes following acute CF induction compared with the baseline. Significant increase in low frequency (LF) expressed in normalized units (n.u.; 5.3 [2.7–15.4] n.u. versus 9.5 [4.1–14.2] n.u., *p* = 0.01) and a low frequency over high-frequency ratio (LF/HF; 0.06 [0.03–0.09] versus 0.10 [0.04–0.17], *p* = 0.008) was observed, whereas the high frequency (HF) significantly decreased (94.7 [91.7–97.3] n.u. versus 90.5 [85.8–95.9] n.u., *p* = 0.01; Figure 3).

### 2.3. Effect of Experimental Acute Heart Failure on Nonlinear Analysis of Heart Rate Variability

The basic parameters from the nonlinear analysis of HRV (SD1, SD2) also prominently and uniformly decreased with acute CF (SD1, 37.8 [7.2–61.7] ms versus 1.4 [1.3–4.4] ms, *p* < 0.001; SD2, 50.8 [21.1–93.8] ms versus 8.3 [3.0–15.0] ms, *p* < 0.001; Figure 4).

## 3. Discussion

The major observation of this study is that acute severe CF significantly influences HRV. To the best of our knowledge, this is the first demonstration that the development of acute CF is associated with a reduction in HRV. Changes in the time-domain and main nonlinear analytic parameters were particularly uniform and pronounced, but changes in frequency-domain parameters were also significant.

It is well known that HFrEF patients have a lower HRV compared with healthy individuals [16]. The significant prognostic value of HRV in the CF population has also been extensively explored [17,18]. Likewise, decreased HRV has been documented in an experimental model of chronic (progressive) CF, primarily induced by rapid right ventricular pacing [19,20]. Surprisingly, the effect of acute CF on HRV has not been described in detail before.

The results of the present study are in agreement with previous experiments that documented the relationship between an acute pathological situation, such as hemorrhagic shock or sepsis, and a decrease in HRV [21,22,23]. In sepsis, the decrease in HRV preceded hemodynamic changes [21,22]. In hemorrhagic shock, hemodynamic and metabolic variables were better discriminants in revealing survivors than HRV metrics [23]. Thus, HRV may not always serve as a marker of the severity of a pathological situation. Nevertheless, in subjects suffering from chronic CF, clinical and veterinary data suggest that HRV negatively correlates with disease severity [24,25]. Our findings support these data and also imply a close relationship between acute severe CF and pronounced autonomic imbalance.

Moreover, according to time-domain and nonlinear HRV parameters, severe acute CF reduces HRV in every single subject. It supports our hypothesis that the patients with high HRV enrolled in neuromodulation studies had insufficiently impaired autonomic balance, signaling a less severe disease. HRV or other autonomic balance estimating tools should be involved in the recruitment process when testing the therapy which affects the autonomic nervous system in CF patients.

Evaluating the effect of our acute CF model on the parasympathetic versus sympathetic nervous system is challenging, as the relationship between both branches of the autonomic nervous system is complex and not simply competitive [26]. Moreover, the interpretation of individual HRV parameters may be problematic, as none of the HRV parameters reflects solely sympathetic nervous system activity change [27]. Nevertheless, a pronounced decrease in time-domain methods, SD1 and HF power signals that a reduction of parasympathetic nervous system activity was the predominant change in the autonomic balance after the development of acute CF in our model [28]. Baroreceptor reflex activation due to loss of myocardial contractility with subsequent systemic hypotension was probably the main vagal activity-modulating factor [29]. Reduction of parasympathetic nervous system activity also probably contributed to a significant decrease in very low frequency (VLF) power [30].

### Study Limitations

Our study has several limitations. Extrapolation of our findings to the clinical setting in HFrEF may be problematic, as we used a model of acute CF, specifically severe acute CF, with features of cardiogenic shock, as opposed to chronic CF. It is, therefore, not clear that changes in HRV would also be as pronounced and uniform in less severe CF.

The hypoxic model of acute CF may be associated with transient hypoxemia in the carotid arteries, beginning after the initiation of hypoxia and lasting until the loss of left ventricular contractility, which is followed by prevalent perfusion of the carotid region by oxygenated blood from the ECMO. During the experiment, decreased cerebral tissue saturation < 40% was detected by near-infrared spectroscopy oximetry in the majority of animals. Despite the fact that we did not observe any severe clinical signs of hypoxic brain damage (seizures or impaired brain stem reflexes) and subcortical structures involved in autonomic nervous system circuits are less susceptible to hypoxia than the cortex, we cannot entirely exclude the influence of short-term hypoxia on the autonomic nervous system. Using standard HRV measures, it is not well possible to differentiate the effect of cerebral hypoxia on HRV from the compensatory HRV reaction due to the loss of myocardial contractility. HRV response can be non-specific regarding the etiology of the insult, and HRV changes are also dependent on the insult severity [31]. Nevertheless, the pattern of HRV change following the development of acute CF was identical in all study subjects, although some experienced minimal, whereas others had longer or deeper cortical hypoxia. Therefore, we believe that changes in HRV predominantly reflect autonomic imbalance caused by acute CF. Moreover, apart from possible transient cerebral hypoxic hypoxia linked to our acute CF model, severe acute CF or cardiogenic shock is frequently accompanied by circulatory cerebral hypoxia. From this aspect, brain hypoxia is an integrative and inseparable part of severe acute CF syndrome and reflects a common clinical scenario. 

HRV frequency-domain parameters may also be impaired by changes in ventilation parameters during the experiment (lower tidal volume and respiratory frequency) [32]. However, time-domain analyses of HRV should not be affected by ventilation [33]. 

Only two 5-min ECG segments were chosen for HRV analysis in each subject (before and after the development of acute CF), with an emphasis on the quality of the traces, as this experimental model is associated with frequent rhythmic disturbances. We did not endeavor to map HRV timeline changes following acute CF induction but aimed to describe the extent of the response.

We expressed HRV according to customary practice in non-indexed values. Nevertheless, there is a negative exponential relationship between heart rate and HRV [34]. As heart rate was significantly higher after the induction of acute CF, when using HRV metrics corrected for heart rate, the change in HRV would be slightly smaller.

Finally, we cannot entirely exclude the influence of sex on HRV, as only female pigs were enrolled due to the easier cannulation of the urinary tract in comparison with boars. In healthy human subjects, significant differences in HRV have been previously described between males and females [35,36]. We may hypothesize that baseline HRV behaves similarly in a porcine model and thus, we cannot exclude that the extent of the changes in HRV would be different in males.

## 4. Materials and Methods

### 4.1. Animal Model

Twenty-four female domestic pigs (*Sus scrofa domestica*, Large White × Landrace crossbreed), 4–5 months of age, with a mean body weight of 45 kg, were enrolled. The animal model used in the present study has been described in detail previously [37,38]. Briefly, after 24 h of fasting, general anesthesia was induced by intramuscular administration of midazolam (0.3 mg/kg) and ketamine hydrochloride (15–20 mg/kg). Following an intravenous bolus of propofol and morphine, endotracheal intubation was performed. General anesthesia was maintained by continuous intravenous administration of propofol, midazolam and morphine. Drop rates of infusions with normal saline and unfractionated heparin were adjusted to maintain a mean central venous pressure of 5–7mmHg and an activated clotting time of 200–250 s, respectively.

### 4.2. Hypoxic Model of Acute Heart Failure

The hypoxic model of severe acute CF used in the present study has been described previously [38,39]. Global myocardial hypoxia was induced by perfusing the anterior body with deoxygenated blood. This was achieved by reducing ventilation support after the initiation of veno-arterial extracorporeal membrane oxygenation (VA-ECMO). In detail, tidal volume was reduced to 100 mL, respiratory frequency to 5 breaths per minute and fraction of inspired oxygen to 0.21. Positive end-expiratory pressure was maintained at 6 cm H_2_O. Ventilation was partially increased to prevent circulatory collapse if needed. Global myocardial hypoxia led to a rapid decrease in ventricular contractility, resulting in reduced ejection fraction, cardiac output and systemic arterial blood pressure. The Hamilton G5 (Hamilton Medical AG, Bonaduz, Switzerland) mechanical ventilator and VA-ECMO in femoro-femoral configuration with a 21 Fr venous cannula, 15–18 Fr arterial cannula and i-cor (Xenios AG, Heilbronn, Germany) or Cardiohelp (Getinge, Rastatt, Germany) systems were used.

### 4.3. Vital Signs and Hemodynamic Monitoring 

Vital signs and invasive hemodynamics were monitored throughout the experiment, including continuous ECG, pulse oximetry measured at the tail, central venous pressure, arterial pressure obtained invasively from the aortic arch using a pigtail catheter and standard fluid-field pressure transducer, pulmonary arterial pressure measured by Swan-Ganz catheter, and cerebral and peripheral tissue oximetry using near-infrared spectroscopy (INVOS, Medtronic, Minneapolis, MN, USA). The Life Scope TR (Nihon Kohden, Tokyo, Japan) and Vigilance II (Edwards Lifescience, Irvine, CA, USA) hemodynamic monitors were used for continuous hemodynamic data recording. Left ventricular performance was assessed by pressure-volume 7 Fr pigtail conductance catheter Scisense (Transonic, Ithaca, NY, USA) inserted into the left ventricle through the aortic valve via the left carotid artery.

### 4.4. Study Protocol and Heart Rate Variability Assessment

ECG was continuously recorded during the experiment. In each subject, two 5-min ECG segments were selected and compared: (i) 5–30 min after insertion of VA-ECMO before induction of myocardial hypoxia and (ii) >60 min after the development of severe CF [40]. Beat-to-beat control of the selected ECG segments was performed to confirm sinus rhythm and avoid irregularities (premature complexes, arrhythmias, artifacts). HRV analysis was automated using LabChart software version 8.1.19 (ADInstruments, Dunedin, New Zealand) in the animal mode for pigs. Time-domain and frequency-domain parameters, as well as nonlinear analytic methods, were used for HRV assessment. In detail, SDNN, RMSSD and pNN50 were chosen as representatives of time-domain measures. HF power, LF power, VLF power expressed in ms^2^, as well as LF/HF and HF and LF bands expressed in normalized units (relative power of particular frequency band related to total power without VLF component), were used to describe the spectral frequency distribution of HRV. In the present porcine model, the HF band was defined as 0.09–2 Hz, the LF band as 0.01–0.09 Hz and the VLF band as <0.01 Hz. Nonlinear analysis was derived from the Poincaré plot that displays the dependence of the RR interval on the preceding RR interval. The plot was fitted by an ellipse, where SD1 represents the standard deviation of the points along the short semi-axis and SD2 the standard deviation of the points along the main semi-axis [22].

### 4.5. Statistical Analyses

Hemodynamic parameters were expressed as mean and standard deviation and analyzed using a two-tailed paired *t*-test. The normality of HRV data was evaluated with the Shapiro-Wilk and D’Agostino-Pearson tests. As all HRV data had non-normal distribution, the Wilcoxon two-tailed matched pairs test was applied and are presented as median and interquartile range. All statistical analyses were performed using GraphPad Prism version 8.3.0 (GraphPad Software, La Jolla, CA, USA). *p*-values < 0.05 were considered statistically significant.

## 5. Conclusions

Acute severe heart failure induced by global myocardial hypoxia is associated with a significant reduction in heart rate variability, which is probably driven predominantly by the reduction of parasympathetic nervous system activity via baroreflex activation. 

## Figures and Tables

**Figure 1 ijms-24-00493-f001:**
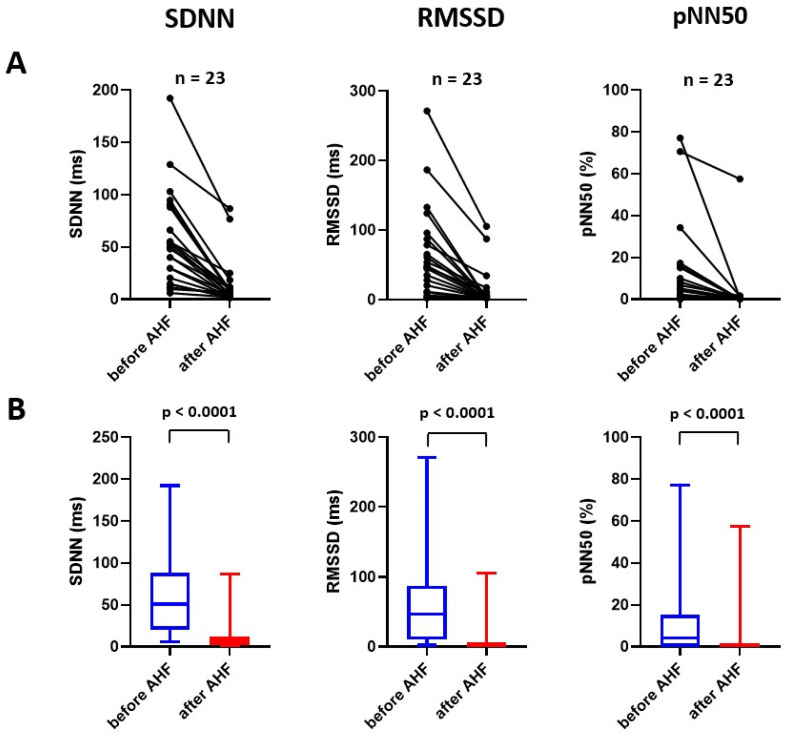
Effect of acute heart failure on time-domain parameters of heart rate variability displayed as an individual graph (**A**) and as median with interquartile range, minimum and maximum value (**B**). AHF = acute heart failure; pNN50 = number of pairs of adjacent NN intervals differing by more than 50 ms divided by the total number of all NN intervals; RMSSD = square root of the mean of the sum of the squares of differences between adjacent NN intervals; SDNN = standard deviation of intervals between normal beats.

**Figure 2 ijms-24-00493-f002:**
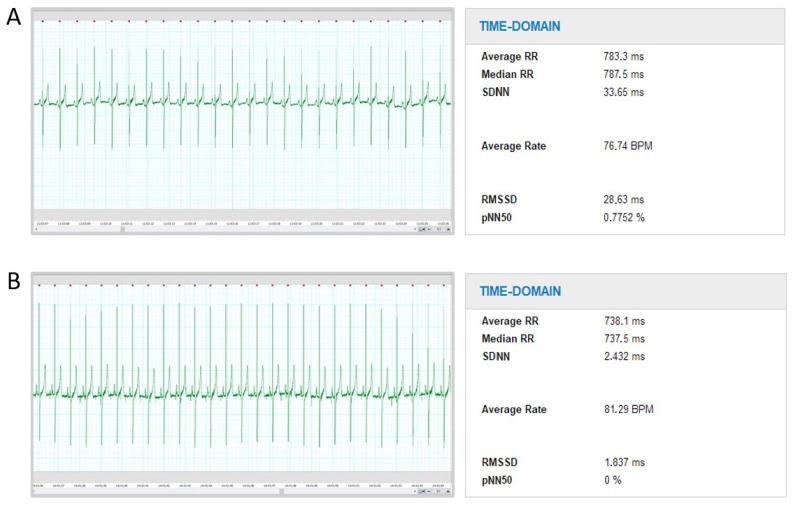
Representative example of HRV analysis comparing time-domain HRV parameters before (**A**) and after (**B**) acute CF induction in one study subject. BPM = beats per minute; pNN50 = number of pairs of adjacent NN intervals differing by more than 50 ms divided by the total number of all NN intervals; RMSSD = square root of the mean of the sum of the squares of differences between adjacent NN intervals; RR = interval between adjacent R waves; SDNN = standard deviation of intervals between normal beats.

**Figure 3 ijms-24-00493-f003:**
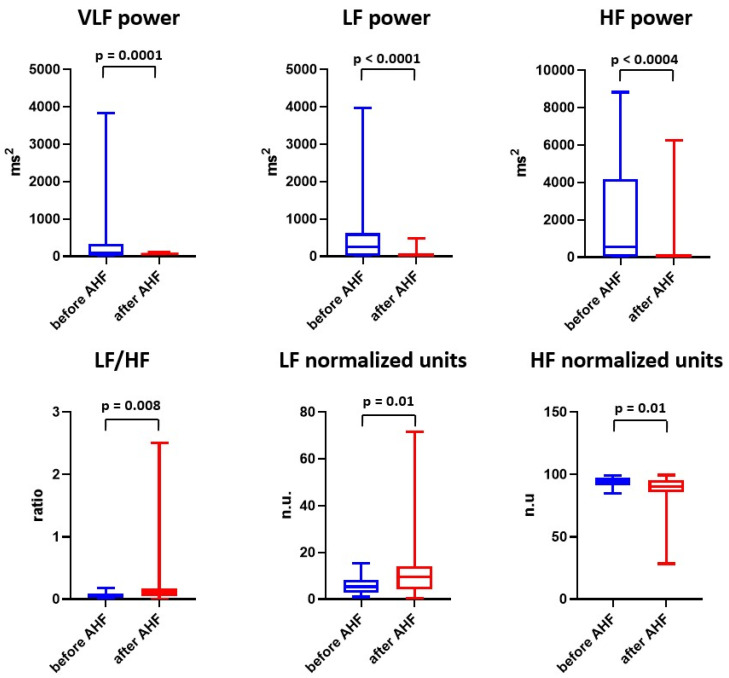
Effect of acute heart failure on frequency-domain parameters of heart rate variability. Data are expressed as median with interquartile range, minimum and maximum value. AHF = acute heart failure; HF = high frequency; LF = low frequency; n.u. = normalized units; VLF = very low frequency.

**Figure 4 ijms-24-00493-f004:**
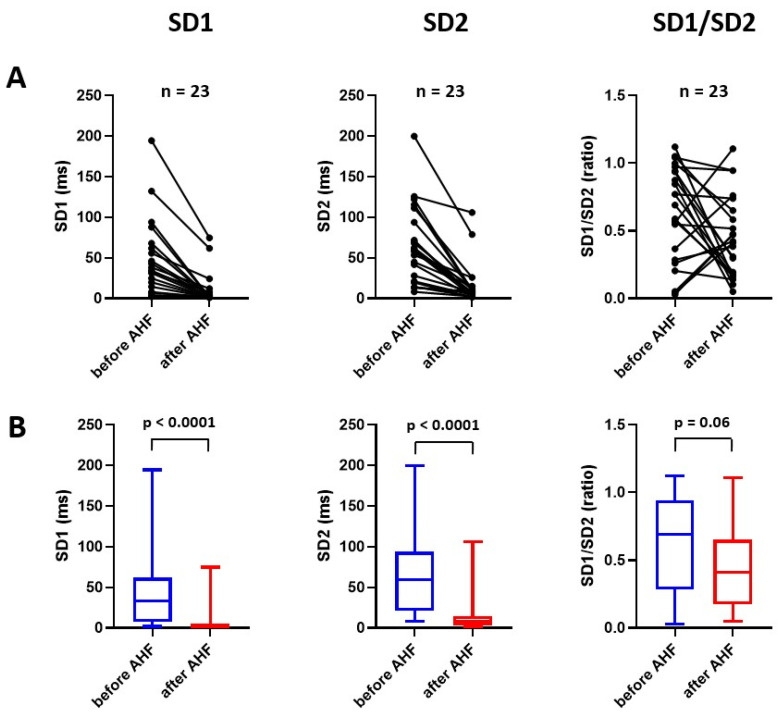
Effect of acute heart failure on nonlinear analysis of heart rate variability displayed as individual graph (**A**) and as median with interquartile range, minimum and maximum value (**B**). AHF = acute heart failure; SD1 = standard deviation of the points along short semi-axis of an ellipse fitted to Poincaré plot; SD2 = standard deviation of the points along long semi-axis of an ellipse fitted to Poincaré plot.

**Table 1 ijms-24-00493-t001:** Main hemodynamic and vital signs variables at baseline and after the induction of acute heart failure.

	Baseline	Acute Heart Failure	*p*-Value
Heart rate (bpm)	88.9 ± 26.0	107.5 ± 26.4	0.01
Mean aortic pressure (mmHg)	94.9 ± 9.3	67.9 ± 23.1	<0.001
Systolic aortic pressure (mmHg)	115.4 ± 12.3	84.1 ± 22.4	<0.001
Diastolic aortic pressure (mmHg)	79.4 ± 8.5	59.9 ± 21.3	0.005
Central venous pressure (mmHg)	3.6 ± 1.5	5.1 ± 2.8	0.03
Mean pulmonary arterial pressure (mmHg)	18.7 ± 4.7	22.8 ± 4.6	0.005
Diastolic pulmonary arterial pressure (mmHg)	13.1 ± 4.6	19.1 ± 4.7	<0.001
Pulse oximetry (tail; %)	98.8 ± 1.6	97 ± 2.4	0.017
Cerebral tissue oximetry (NIRS; %)	62.3 ± 6.7	53.2 ± 14.5	0.046
Hind limb tissue oximetry (NIRS; %)	56.5 ± 7.3	54.3 ± 7.2	0.051
Left ventricular dP/dt_max_ (mmHg/s)	1740 ± 454	949 ± 316	<0.001
Left ventricular end-diastolic volume (mL)	126.9 ± 11.1	135.8 ± 15.9	0.058
Stroke volume (mL)	74.8 ± 9.8	23.8 ± 7.4	<0.001
Cardiac output (L/min)	7.2 ± 1.8	2.6 ± 0.7	<0.001
Left ventricular ejection fraction (%)	57.9 ± 4.9	17.2 ± 5.0	<0.001

Data are expressed as mean and standard deviation. dP/dt_max_ = maximal rate of pressure rise during ventricular contraction; NIRS = near-infrared spectroscopy.

## Data Availability

The data regarding the presented study are available from the corresponding author upon reasonable request.

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
