# Peer review of "Acute Severe Heart Failure Reduces Heart Rate Variability: An Experimental Study in a Porcine Model"

_ijms, 2022, doi:10.3390/ijms24010493_

Round 1

Reviewer 1 Report

Summary:

The main aim of this study is to evaluate and test the hypothesis that, acute severe heart failure immediately influences heart rate variability. The authors conducted a study that included 23 subjects who are eligible for heart rate variability analysis. The authors analyzed the effect of acute heart failure on time-domain  and frequency domain parameters of heart rate variability, non-linear analysis of heart rate variability.

Comments:

1.     This study has several limitations.

2.     Could the authors explain the reason for including only female pigs for this study?

3.     Corrections in the English are required.

Reviewer 2 Report

Dear Sir/Madam,

I had the opportunity to act as a reviewer on the recent submission by Naar et al. to the International Journal of Molecular Sciences.

The authors present an interesting paper studying the heart rate variability in an experimental study on pigs with acute severe heart failure. The key finding is that acute severe heart failure reduces heart rate variability. 

The manuscript is very well structured and written. However, some issues need to be addressed: 

1.     Materials and Methods are usually presented after introduction. I recommend switching Results and Materials and Methods sections. 

2.     The Introduction does not offer any information on acute heart failure and HRV, I strongly recommend adding this data.

3.     What is the translational value of this study? How does it translate into clinical practice?

Best regards,

Round 2

Reviewer 2 Report

Dear Sir/Madam,

Thank you for reviewing the manuscript and addressing the mentioned issues. These were adequately answered. Therefore, the manuscript seems suitable for publishing in the present form.

Best regards

Author Response

The authors thank the reviewer for favorable comments and positive recommendation.

Reviewer 3 Report

Dear Authors, 

The manuscript has been improved; however, the explanation of HRV parameters is incorrect. A comprehensive study was carried out, but the result seems weak, as no specific conclusions were drawn. The pathophysiology of severe cardiogenic shock includes autonomic dysfunction due to cerebrovascular hypoxia. However, the method used does not allow separate assessment of autonomic function associated with heart failure compensation and autonomic dysfunction caused by hypoxic brain damage.

HRV parameters in the time domain usually correspond to parasympathetic activity. Their low values are due to low cardiac output and, as a result, low baroreflex activation. This aspect was not discussed.

In my opinion, low-frequency parameters (TP, HF, LF) correspond to autonomic neurogenic deficit due to cerebral circulatory hypoxia.

The Authors should discuss in more detail the origin of the modification of HRV parameters.

Round 3

Reviewer 3 Report

Dear Authors,

I believe that the manuscript has been improved enough to warrant publication in the IJMS. All methodological flaws and limitations of the study are explained.